# Highly Thermal Conductive Graphite Films Derived from the Graphitization of Chemically Imidized Polyimide Films

**DOI:** 10.3390/nano12030367

**Published:** 2022-01-24

**Authors:** Meijiao Sun, Xiaoqiang Wang, Zhengyu Ye, Xiaodong Chen, Yuhua Xue, Guangzhi Yang

**Affiliations:** 1School of Materials and Chemistry, University of Shanghai for Science and Technology, Shanghai 200093, China; 18307416856@163.com (M.S.); XQ320557@163.com (X.W.); 1935040329@st.usst.edu.cn (Z.Y.); xueyuehua@usst.edu.cn (Y.X.); 2Taihu Jinzhang Science & Technology (Anhui) Co., Ltd., Anqing 246000, China; chenxd@jzt3.com

**Keywords:** polyimide film, chemical imidization, graphitization, thermal conductivity

## Abstract

With the large-scale application and high-speed operation of electronic equipment, the thermal diffusion problem presents an increasing requirement for effective heat dissipation materials. Herein, high thermal conductive graphite films were fabricated via the graphitization of polyimide (PI) films with different amounts of chemical catalytic reagent. The results showed that chemically imidized PI (CIPI) films exhibit a higher tensile strength, thermal stability, and imidization degree than that of purely thermally imidized PI (TIPI) films. The graphite films derived from CIPI films present a more complete crystal orientation and ordered arrangement. With only 0.72% chemical catalytic reagent, the graphitized CIPI film achieved a high thermal conductivity of 1767 W·m^−1^·K^−1^, which is much higher than that of graphited TIPI film (1331 W·m^−1^·K^−1^), with an increase of 32.8%. The high thermal conductivity is attributed to the large in-plane crystallite size and high crystal integrity. It is believed that the chemical imidization method prioritizes the preparation of high-quality PI films and helps graphite films achieve an excellent performance.

## 1. Introduction

With the rapid development of electronic devices, the fast speed and high frequency of signal transmission inevitably cause a large amount of heat release at high thermal density, which generates a high requirement for efficient thermal conductivity materials within these electronic devices [1,2]. In recent years, carbon materials with a variety of allotropes have attracted much attention in the field of heat management due to their remarkable thermal conductivity (λ) properties in comparison with other materials [3,4]. There are many kinds of available graphited carbon materials used for heat transfer, such as graphite paper (λ = 200–500 W·m^−1^·K^−1^) [1], graphite film derived from perforated graphite sheet (λ = 179 W·m^−1^·K^−^^1^) [5], ultra-thick graphene film (λ = 1204 W·m^−1^·K^−^^1^) [6], carbon nanotube films (λ = 766 W·m^−1^·K^−^^1^) [7], highly oriented pyrolytic graphite (HOPG) (λ = 1950 W·m^−1^·K^−1^) [8], graphite films/blocks derived from some polymers [9,10], flexible graphite sheets (λ = 575 W·m^−1^·K^−1^) [11], and single-layer graphene film (λ = 5300 W·m^−1^·K^−^^1^) [12,13]. In general, the thermal performance of carbon materials is closely related to their structure at the nanometer scale, and phonon thermal transport [14]. In addition, some research has also focused on multiplex structural graphite materials, such as graphitized reduced graphene oxide/polyimide films (λ = 1467 W·m^−1^·K^−^^1^) [15], graphene/carbon fiber (λ = 977 W·m^−1^·K^−^^1^) [16], dual-functional graphene/carbon nanotube thick film (λ = 933 W·m^−1^·K^−^^1^) [17], graphene/graphitized polydopamine/carbon nanotube all-carbon ternary composites films (λ = 1597 W·m^−1^·K^−^^1^) [18] and so on. However, it is difficult to achieve the simplified preparation process and required thickness to ensure excellent thermal conductivity. At present, graphite film derived from the pyrolysis of polymer film is considered to be extremely promising due to its excellent performances and controllable preparation process and shapes [9,19], such as polyimide [20] and polyacrylonitrile [21].

Aromatic polyimide (PI) film has attracted much attention in scientific and industrial research with the priorities of convenient processability, easy graphitization, high carbon yield, and scale production [20,22]. Meanwhile, the molecular structure and nature of precursor PI films have a significant effect on the structure of the resultant carbon and graphite films. Although graphite films derived from various PI films of the Kapton [23], Upilex [24], and Novax [25] types by traditional graphitization treatment have been systematically studied, there are still many complicated behaviors during carbonization and graphitization in the films which require further study [19]. During the heating process, the changes in the size and thickness of PI film created cracks, wrinkles, and fragments [22]. In order to obtain a complete and high orientation structure of graphite film with outstanding thermal conductivity, the composition and characteristics of the monomer structure of the starting PI, as well as the pyrolytic graphitization process, are particularly important [26]. Some studies have focused on the catalytic carbonization and graphitization of PI film by the introduction of nickel [27], SiC [28], g-C_3_N_4_ [2], and pyridine [29]. Other research has managed to improve the thermal conductivity of precursor PI film via the addition of thermally conductive fillers, such as BN [30] and CN nanosheets [31]. Unexpectedly, there exist defects and turbostratic phases in the cross-sections of graphitized films and phase separation due to the addition of fillers. The structural transformation of graphited PI film with high thermal conductivity largely depends on the orderly structure of the precursor’s PI molecular chain during the carbonization and graphitization stage [32].

An effective measure to improve the quality of the precursor PI film with high imidization degree (ID) and in-plane orientation is to select a suitable imidization method. The relation between the ID and properties of PI films prepared by chemical imidization and thermal imidization methods was investigated by Wang et al. [33]. It was found that the high ID and in-plane orientation of the PI molecular chain greatly increased the tensile strength and modulus, which may promote the orientation arrangement of structure during the graphitization of precursor film. Furthermore, the graphite structure of the in-plane orientation is beneficial to interfacial heat transmission between phonons, resulting in an improvement in thermal conductivity [34,35].

In this work, several amounts of chemical dehydration reagent are used as catalysts in the imidization reaction of polyamide acid to prepare precursor PI films with different IDs and aggregation structures. Graphite films are derived from the graphitization of the obtained PI films at a high temperature of 2800 °C. The heat diffusion behaviors and catalytic effect of chemical dehydration reagent on the graphitization of PI films are investigated. The crystalline integrity and defect repair of the graphite structure are analyzed in detail by Raman and X-ray diffraction. The prepared graphite films have a directional layered structure with high compactness and exhibit a superior heat diffusion property.

## 2. Experimental Section

### 2.1. Preparation of Graphite Film

Figure 1 represents the preparation process of graphite films. Pyromellitic dianhydride (PMDA) and 4,4′-diaminodiphenyl ether (ODA) were dissolved in *N*,*N*-Dimethylacetamide (DMAc) solvent to prepare polyamide acid (PAA) solution with a solid weight content of 14%. Different amounts of chemical catalytic reagent (phosphorus compound supplied by Taihu Jinzhang Science & Technology (Anhui) Crop., Ltd., Anqing, China) were added to the PAA solution (by 0.5–0.8% weight content) and stirred evenly at the room temperature. The PAA solution was then defoamed under vacuum, coated for film on a glass plate and imidized at 300 °C for 70 min. In the experiment, PI films with different weight percents of imide reagent were prepared, which were 0%, 0.5%, 0.56%, 0.64%, 0.72% and 0.8%, respectively. A percentage of 0 means that no imide reagent was used and regarded as a thermal imidization. The samples are named as TIPI and CIPI-w (w is the weight percent of imide reagent), respectively.

PI films were carbonized at 1000 °C for 30 min in N_2_ atmosphere and the carbonized PI films are named as c-TIPI and c-CIPI-w (w is the weight percent of imide reagent), respectively. The graphitization of PI carbon films was carried out at 2800 °C for 30 min in Ar atmosphere. The obtained graphited PI films are named as g-TIPI and g-CIPI-w (w is the weight percent of imide reagent), respectively.

### 2.2. Characterization of Materials

The chemical structure, crystallization degree, stress–strain behavior, thermal properties, and glass transition temperature (Tg) of the films were analyzed by Fourier transform infrared spectra (FTIR, Perkin Elmer, Waltham, MA, USA), X-ray diffraction (XRD, Bruker, Bremen, Germany), Materials Testing Machine (ZwickRoell, Ulm, Germany) (each sample was measured at least 3 times), thermogravimetric analysis (TG, Perkin Elmer), and differential scanning calorimetry (DSC, Netzsch, Hanau, Germany), respectively. The structure of the carbonized and graphited films was characterized by Raman spectra (Horiba, Lille, France). The surface and cross-section morphology of films were observed by scanning electron microscope (SEM, FEI Quanta FEG, FEI, Hillsboro, OR, USA) with an accelerating voltage of 20 kV.

### 2.3. Measurement of Thermal Conductivity

The graphitized PI films were rolled by Roller tablet press and drilled into a disc shape for measurement of thermal diffusivity (α, mm^2^·s^−1^) by a laser flash analyzer (LFA467 Nanoflash, Netzsch). During the measurement, each graphite film was automatically measured 3 times and an average value was obtained. The error for α is less than ± 15 mm^2^·s^−1^. The thermal conductivity (λ, W·m^−1^·K^−1^) is calculated by Equation (1) according to the literature [36].
(1)$$\lambda =\rho \cdot C_{p}\cdot \alpha$$
where ρ represents the density (g·cm^−3^, obtained according to $\rho =m\cdot V^{-1}$, where m and V are the mass and volume of the sample, respectively) and $C_{p}$ represents the specific heat capacity (J·g^−1^·K^−1^).

## 3. Results and Discussion

### 3.1. Structure and Properties of PI Films

Figure 2 represents the possible mechanism of the changes in the PAA molecular chain during chemical imidization and thermal processes. Compared with thermal imidization, chemical imidization is already partially imidized with orientation structure during the solvent evaporation at the low-temperature stage [33,37]. In the high-temperature imidization stage, the unimidized PAA molecular chain surrounded the formed PI molecular chain and transformed into an in-plane orientation arrangement due to the plasticization of the residual solvent molecules.

The FTIR characterization of the prepared PI films is shown in Figure 3a. It can be observed that the peaks at approximately 1780 cm^−1^ and 1720 cm^−1^ correspond to the symmetric and asymmetric stretching vibrations of the imide ring C = O bond, respectively. The peak at 1360 cm^−1^ is assigned to the stretching vibrations of the C-N bond. The deformation vibration peak of the imide ring is found at 725 cm^−1^. All characteristic peaks indicate that the prepared sample is a polyimide molecular chain structure. The intensity ratio of the peak at 1360 cm^−1^ of C-N stretching vibration (quantifying ID) on the imide ring to the peak at 1500 cm^−1^ (as internal standard) was used to calculate the ID of PI films; it is expressed as the following equation [38]:(2)$$\mathrm{ID}=\frac{A_{1360{\mathrm{cm}}^{-1}}}{A_{1500{\mathrm{cm}}^{-1}}}$$
where $A_{1360{\mathrm{cm}}^{-1}}$ is the absorbance value 1360 cm^−1^ peak and $A_{1500{\mathrm{cm}}^{-1}}$ is the absorbance value of the C-C stretching vibration on the benzene ring.

Obviously, the ID values of all CIPI-w films are higher than those of TIPI films, indicating that the chemical catalytic reagent promotes the imidizaiton of PAA to increase ID at the solvent evaporation stage. The CIPI-0.72% film presents a higher ID value of 87.5% in comparison with other CIPI-w and TIPI (84.9%) films. With an increase in the dehydration reagent, exceeding a certain amount, the ID value begins to decrease, possibly due to the decline of acidity [39]. This result shows that an appropriate amount of chemical dehydration reagent has the best catalytic imidization effect. The aggregation structure of the PI films was analyzed by the X-ray diffraction (XRD), as shown in Figure 3b. A wide diffraction peak appears at 10–20° in the XRD patterns for TIPI and CIPI-w films, indicating a typical amorphous structure [40]. The typical characteristic peak is located at 13.7°, which is indexed to the corresponding (101) plane of PI (PDF#54-2400). The highest peak intensity of CIPI-0.72% may be attributed to the best aggregation structure formed in the imidization process.

Subsequently, the mechanical properties, thermal resistance, and thermal stability of the TIPI and CIPI-w films were measured as shown in Figure 4a–d. The results of ID, tensile strength, T_5,_ and T_g_ are summarized in Table 1. In Figure 4a, it can be seen that the tensile strengths of all CIPI-w films are significantly higher than that of TIPI film (113.22 MPa). This may be due to the gradually formed orientation and crystallization, and the enhancement of the molecular chain force under the catalysis of chemical dehydration reagent. With the addition of chemical dehydration reagent from 0.5% to 0.8%, the tensile strength values were 119.44 MPa, 119.26 MPa, 119.09 MPa, 122.38 MPa, and 121.73 MPa, respectively. Meanwhile, the relationship between tensile strength and ID are characterized in Figure 4b; the changes in the tensile strength and the ID of PI films with different chemical catalytic reagents show a consistent trend. The index of ID strongly correlates with the degree of in-plane orientation, mechanical strength, and thermal properties of PI film [41]. It can be concluded that the high tensile strength of PI films is attributed to the high imide degree and ordered aggregation phase. The thermal stability of PI films was analyzed by TG (accuracy ± 0.1 °C) and DTG, which are presented in Figure 4c. It has been found that almost no weight loss was detected before 500 °C. The temperature of 5% weight loss (T_5_) for CIPI-w films exceeds 550 °C, which is higher than that of TIPI film (524 °C). The T_5_ values of all CIPI-w films are presented in Table 1. The temperature of the maximum thermal decomposition of PI films is approximately 600 °C according to the DTG curves. This indicates that the strong molecular chain forces of CIPI-w films with high ID and in-plane orientation enhance the thermal stability during the high-temperature pyrolysis process. The glass transition temperature (T_g_) of PI films was measured by DSC (the rate of heating was at 10 °C/min with N_2_ flow rate at 50 mL/min, accuracy ± 0.1 °C), as shown in Figure 4d. With the increase in chemical catalytic reagent from 0.5% to 0.8%, the T_g_ values of CIPI films are 388.2 °C, 389.1 °C, 390.3 °C, 390.9 °C and 391.2 °C, respectively; these values are higher than that of 369.6 °C for TIPI film. This may be attributed to the increased intermolecular forces caused by the orderly aggregated structure under the chemical catalytic reagent.

### 3.2. Morphology and Structure of Graphited PI Films

Figure 5a–f show the surface and cross-sectional morphology of TIPI and CIPI-0.72% films after graphitization, respectively. In Figure 5a–c, there are faults, and cracks that could be found on the surface of the g-TIPI film. Disordered layer arrangement and gaps between the graphite flakes could be seen in the cross-section of graphite films. Comparatively, for the g-CIPI-0.72% film, the surface exhibits a good flatness, and the cross-section is dense with a layered structure in the planner direction, which is consistent with the results reported elsewhere [22,42]. As seen in Figure 5g–i, the graphitized PI films display a dark grey color, intact appearance, and flexible adaptability, which can be folded or cut into required shapes.

The XRD and Raman characterizations of PI films after carbonization and graphitization are shown in Figure 6a–c. The peak at 13.7° of (101) planes for PI film disappears, while the (002) and (004) diffraction peaks emerge at 26.5° and 54°, corresponding to the typical graphite structure of carbon materials [1]. Compared with g-TIPI film, the peak intensity of g-CIPI-w film is largely increased, indicating a good crystallinity and orientation of graphite sheets. For g-TIPI film, 2θ is at 26.48°, and the corresponding interlayer distance is 0.3363 nm. However, the 2θ is 26.54° for the g-CIPI-0.72% film, corresponding to the d_(002)_ of 0.3355 nm, which is close to the ideal layer spacing of graphite lattice (0.3354 nm) [43]. This result suggests that the amorphous structure of PI transforms into graphite crystalline structure after graphitization. It is reported that PI films with good tensile strengths have a facilitation effect on graphitization to obtain better directional crystallization [44]. Combined with Raman spectroscopy analysis, for carbonized PI film, the strong D band (centered at 1352.04 cm^−1^) is ascribed to the disordered or defective structure of SP^3^ carbon atoms, while the G band (centered at 1596.39 cm^−1^) represents the ordered SP^2^ carbon atoms. In general, the intensity ratio of peak D to peak G (I_D_/I_G_) is used to evaluate the structural defects and graphitization process of materials [45]. Before graphitization, the I_D_/I_G_ values of carbonized PI films with an increasing chemical imide reagent from 0% to 0.8% are calculated as 0.956, 0.974, 0.955, 0.981, 0.985 and 0.993, respectively. After high-temperature graphitization, the intensity of the D band peak is much weaker than that of the G band peak for all graphite films, with the lowest I_D_/I_G_ value of 0.059 for g-CIPI-0.72%, indicating that during the graphitization process the structural defects are repaired and the crystallization is greatly increased [42]. Meanwhile, an increase in the corresponding in-plane crystallite size (La) is found, which was calculated according to Cançado’s equation [46]. Table 2 summarizes the La values of each graphite film; it can be seen that the La values of g-CIPI-w films are 192.17 nm, 319.57 nm, 314.82 nm, 324.25 nm and 286.88 nm, respectively. The g-CIPI-0.72% film shows the largest La of 324.25 nm, compared to that of g-TIPI (123.21 nm). It shows that the more oriented aggregation structure of the precursors promotes the relatively complete repair of lattice defects and form a large grain size after high-temperature graphitization process.

The Lorentz fitting curve of 2D peaks is shown in Figure 6d. The asymmetric shape of the 2D band is created using a disordered layer structure ($G_{2D}^{\prime }$, ~2700 cm^−1^) and graphite AB stacking oriented structure ($G_{3\mathrm{DA}}^{\prime }$ and $G_{3\mathrm{DB}}^{\prime }$, respectively, at 2680 cm^−1^ and 2725 cm^−1^) [47]. However, compared with g-TIPI, the 2D peak of g-CIPI-w is evidently increased. The crystal integrity of the graphite structure (R) can be calculated by Equation (3) to analyze the film orientation and structural integrity [48].
(3)$$R=\frac{I_{G_{3\mathrm{DB}}^{\prime }}}{I_{G_{3\mathrm{DB}}^{\prime }}+I_{G_{2D}^{\prime }}}$$
where $I_{G_{2D}^{\prime }}$ and $I_{G_{3\mathrm{DB}}^{\prime }}$ is the intensity of $G_{2D}^{\prime }$ and $G_{3\mathrm{DB}}^{\prime }$ peaks of graphited TIPI and CIPI-w films, respectively, and the calculated R value corresponds to the structural integrity of film. The peak intensity values of each peak of 2D peak fitting are listed in Table 2. The calculated R values of each g-CIPI-w films are 0.892, 0.919, 0.863, 0.928 and 0.915, respectively, which are higher than that of g-TIPI film (0.842). The R value reaches the maximum of 0.928 for g-CIPI-0.72%, which represents the higher orientation degree and the better crystal integrity. Figure 6e shows the change of I_D_/I_G_ values along with R for graphitized PI films. It is found that there are fewer structural defects and a better crystal integrity for the graphite films derived from the graphitization of the CIPI films in comparison with the graphited TIPI film. This may be due to the good orientation and crystallinity of the precursor PI film under the chemical catalytic of imidization reagent.

### 3.3. Thermal Conductivity of Graphited PI Films

The thermal conductivity of g-TIPI and g-CIPI-w films is calculated and shown in Figure 7a. It is known that thermal conductivity values are calculated by thermal diffusivity (α) ∗ density ∗ specific heat capacity ($C_{p}$) according to Equation (1). The thermal diffusivity of each graphited PI film is presented by each point in the red lines of Figure 7a. A laser flash analysis (LFA467 NanoFlash) was implemented to measure the thermal diffusivity of graphited PI film (the error for α is less than ±15 mm^2^·s^−1^). The density of each sample is listed in Appendix A. The specific heat capacity of the graphite film was obtained through differential scanning calorimetry from the literature (an average $C_{p}$ of 0.756 J·g^−1^·K^−1^) [49]. The addition of chemical catalytic reagent makes the thermal conductivity of graphited CIPI-w films higher than that of graphited TIPI film. With the increase in the layer-oriented structure, the defect concentrations decrease, which are conducive to the improvement of thermal conductivity [50]. When the chemical catalytic reagent ratio is lower than 0.72%, a low density (less than 1.9 g·cm^−1^) and poor thermal diffusivity are shown due to incomplete graphitization and defects. It can be seen that the g-CIPI-0.72% film reaches the highest thermal conductivity of 1767.2 W·m^−1^·K^−1^ (the density and α is 2.1 g·cm^−1^ and 1113.1 mm^2^·s^−1^, respectively), increased by 32.8% in comparison with the g-TIPI film (1330.5 W·m^−1^·K^−1^). According to the Raman test, the relationship between I_D_/I_G_, La and thermal conductivity are analyzed. As shown in Figure 7b, excellent thermal conductivity of graphite film is benefited by good structural orientation, large crystallite size, and better repairing of structural defects. Table 3 lists the comparison of thermal conductivity properties of multiple graphite films after graphitization reported in the literature, indicating that the high-orientation graphite film obtained by graphitizing CIPI film has an outstanding thermal conductivity.

The excellent thermal conductivity enables graphited PI films to have a broad range of technological applications, such as LED lamps and smart electronic equipment. In this work, the g-CIPI-0.72% and g-TIPI films as thermal management materials were attached to the bottom of LED lamps with the working voltage of 26 V and were evaluated by an infrared thermal imager (UQ-AAA, Seek thermal) to monitor the surface temperature of the samples with time. The schematic diagram for heat source test equipment is shown in Figure 8a. The surface temperature of g-CIPI-0.72% and g-TIPI films with time was monitored by an infrared thermal imager, and the corresponding infrared thermal images at different times of LED operations are shown in Figure 8b,c. Compared with g-TIPI film, the g-CIPI-0.72% film exhibits faster heating rate increases, which is caused by the fast heat dissipation graphite film with high thermal conductivity [31]. In addition, the surface temperature of graphited film can quickly decrease to a certain temperature. When the LED light was turned on, compared with g-TIPI film, g-CIPI-0.72% film had a high and uniform distribution, which revealed a faster heat dissipation efficiency (for example, at 180 s, the temperature difference between g-TIPI and g-CIPI-0.72% films reached 3 °C).

## 4. Conclusions

In this study, we comparatively analyzed the imidization degree and thermal conductivity properties of PI films and graphite films prepared by chemical and thermal imidization methods. The chemical catalyst reagent was introduced to speed up the imidization reaction and promote the orderly stacking of the molecular chain. The well aggregated structure and high imidization degree for precursor PI film is conducive to forming a graphite structure of an ordered interlayer arrangement, large crystallite size, and small lattice defect, thus improving the thermal conductivity of graphite film. The excellent thermal conductive property of chemically imidized polyimide-derived graphite films gives it great potential for heat diffusion and thermal management.

## Figures and Tables

**Figure 1 nanomaterials-12-00367-f001:**
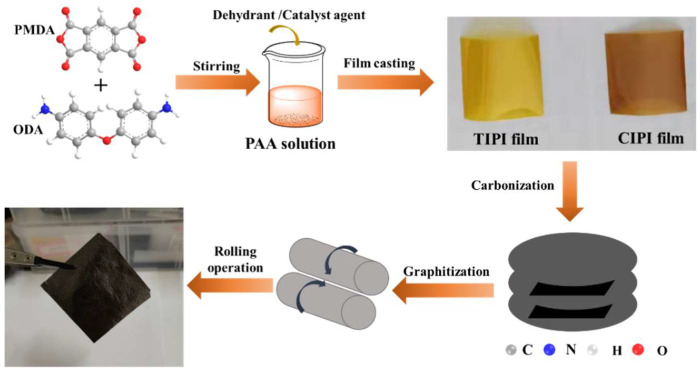
Preparation process of graphite films.

**Figure 2 nanomaterials-12-00367-f002:**
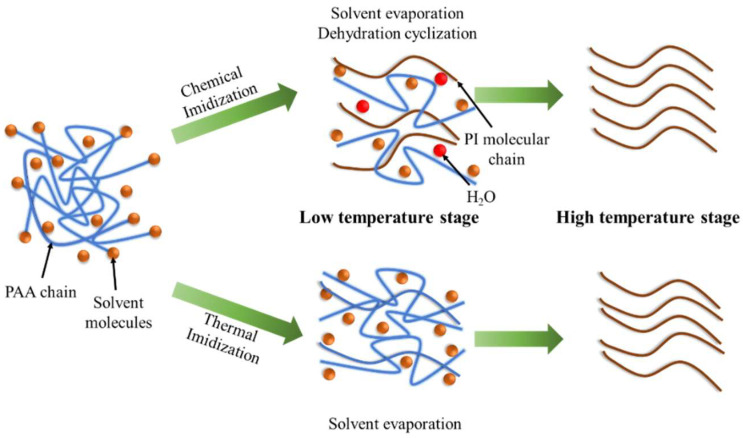
Mechanism diagram of chemical and thermal imidization of PAA.

**Figure 3 nanomaterials-12-00367-f003:**
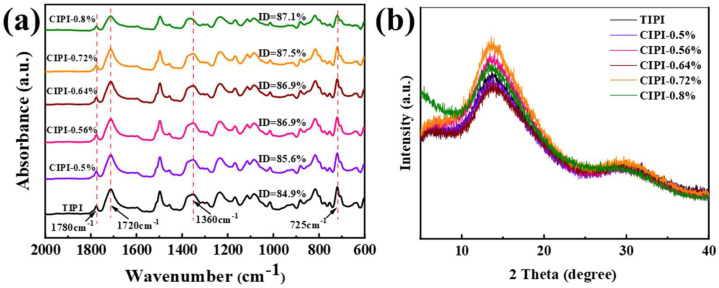
Structure analysis of TIPI and CIPI-w films: (**a**) FTIR; (**b**) XRD.

**Figure 4 nanomaterials-12-00367-f004:**
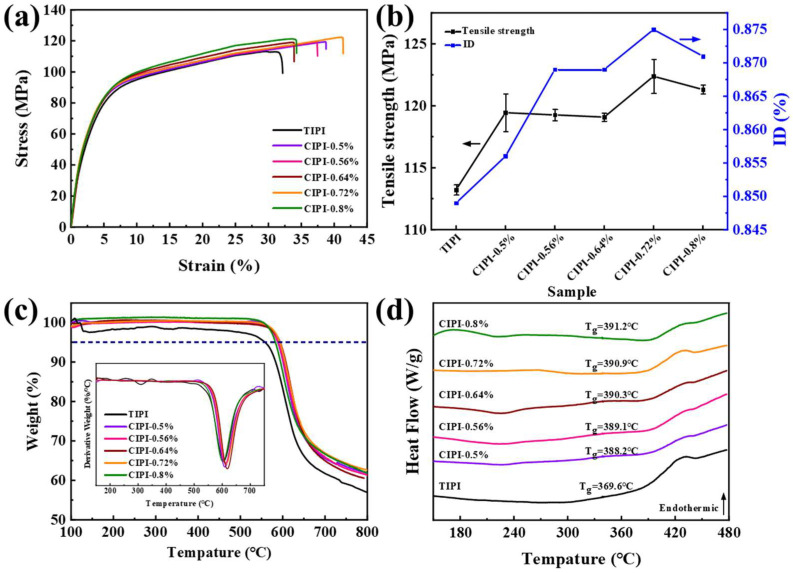
Mechanical and thermal properties of TIPI and CIPI-w films: (**a**) stress–strain curves; (**b**) the relationship between tensile strength and ID; (**c**) TG and DTG; (**d**) DSC.

**Figure 5 nanomaterials-12-00367-f005:**
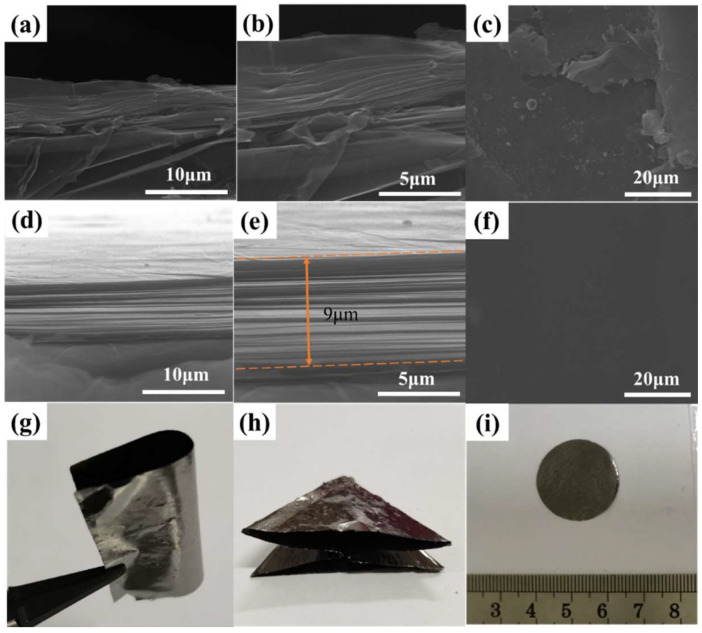
(**a**,**b**) SEM images of g-TIPI film cross-section; (**c**) surface images of g-TIPI film; (**d**,**e**) SEM images of g-CIPI-0.72% film cross-section; (**f**) surface images of g-CIPI-0.72% film; (**g**–**i**) optical photograph of graphited films.

**Figure 6 nanomaterials-12-00367-f006:**
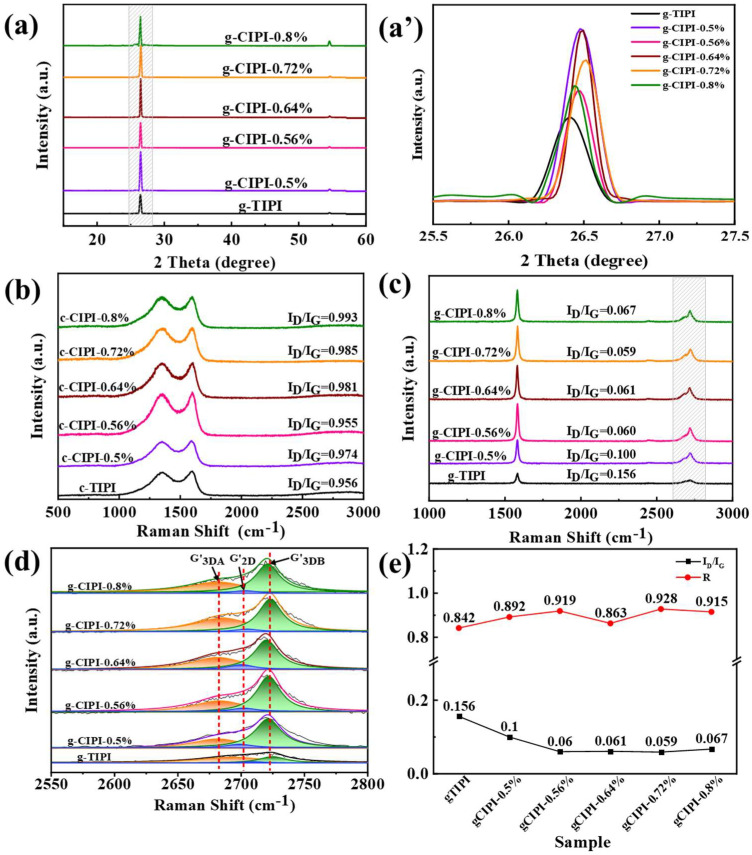
(**a**,**a’**) XRD patterns of g-TIPI and g-CIPI-w films; Raman spectroscopy: (**b**) c-TIPI and c-CIPI-w films, (**c**) g-TIPI and g-CIPI films, and (**d**) Lorentz fitting curve of Raman 2D peak; (**e**) I_D_/I_G_ and R value.

**Figure 7 nanomaterials-12-00367-f007:**
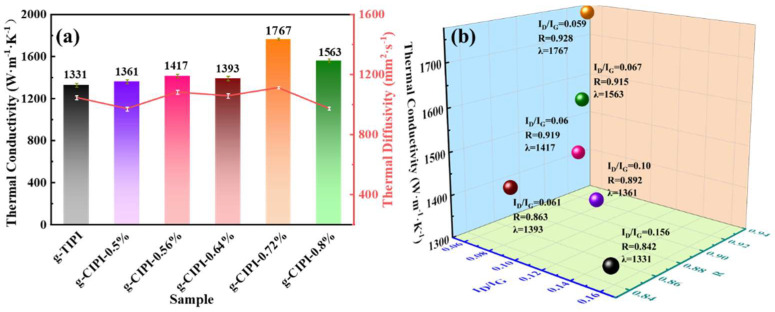
(**a**) The thermal properties measured of g-TIPI and g-CIPI-w films; (**b**) scatter diagram of thermal conductivity.

**Figure 8 nanomaterials-12-00367-f008:**
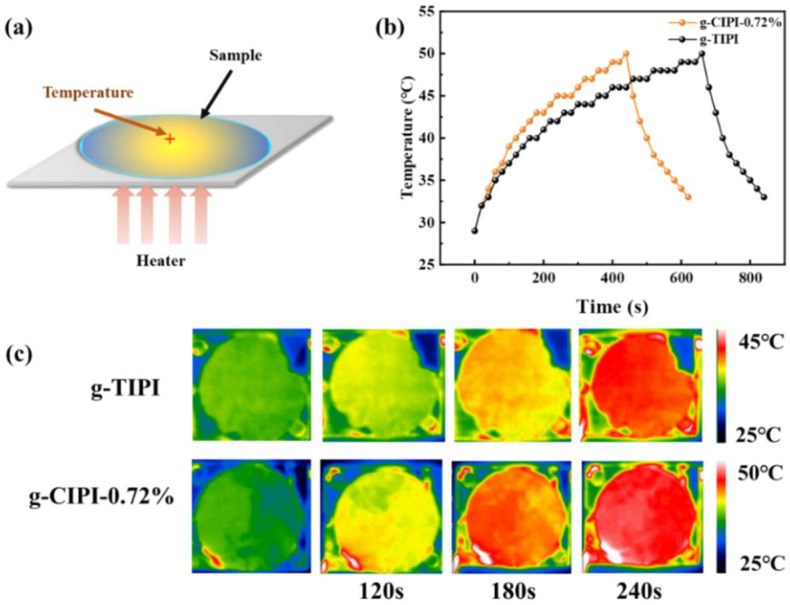
(**a**) The schematic diagram for heat source test equipment; (**b**) the relationship curves of temperatures of g-TIPI and g-CIPI-0.72% films with time; (**c**) infrared thermal images of g-TIPI and g-CIPI-0.72% films.

**Table 1 nanomaterials-12-00367-t001:** ID, mechanical and thermal performances of TIPI and CIPI-w films.

Sample	ID/%	Tensile Strength/MPa	T_5_/°C	T_g_/°C
TIPI	84.9	113.22	524.0	369.6
CIPI-0.5%	85.6	119.44	559.3	388.2
CIPI-0.56%	86.9	119.26	589.0	389.1
CIPI-0.64%	86.9	119.09	593.7	390.3
CIPI-0.72%	87.5	122.38	593.0	390.9
CIPI-0.8%	87.1	121.73	561.7	391.2

**Table 2 nanomaterials-12-00367-t002:** Two-dimensopmal peak fitting, R, I_D_/I_G_ and La values of g-TIPI and g-CIPI films.

	2D Peak Fitting				I_D_/I_G_	La (nm)
	$I_{G_{3DA}^{\prime }}$ (a.u.)	$I_{G_{3DB}^{\prime }}$ (a.u.)	$I_{G_{2D}^{\prime }}$ (a.u.)	R		
g-TIPI	30.65	33.89	6.36	0.842	0.156	123.21
g-CIPI-0.5%	51.73	154.41	18.6	0.892	0.1	192.17
g-CIPI-0.56%	62.53	191.23	16.85	0.919	0.06	319.57
g-CIPI-0.64%	68.12	156.88	24.94	0.863	0.061	314.82
g-CIPI-0.72%	72.56	172.97	13.39	0.928	0.059	324.25
g-CIPI-0.8%	55.04	151.39	13.99	0.915	0.067	286.88

**Table 3 nanomaterials-12-00367-t003:** Comparison of thermal conductivities of graphite films.

Base Material	Filler/Catalyst	Graphitization Temperature/°C	Thermal Conductivity/(W/m·K)	Ref
GO	PDA	3000	1584	[36]
PMDA/ODA	-	3200	1950	[22]
PAN	GO	2800	1282	[21]
GF	-	2850	1204	[6]
GE		2850	1529	[51]
PMDA/ODA	Propionic anhydride/Pyridine	2950	1092	[29]
GO	-	2750	1200	[52]
PMDA/ODA	Phosphorus compound	2800	1767	This work

## Data Availability

The data presented in this study are available on request from the corresponding author.

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
