# Peer review of "Highly Thermal Conductive Graphite Films Derived from the Graphitization of Chemically Imidized Polyimide Films"

_nanomaterials, 2022, doi:10.3390/nano12030367_

Round 1

Reviewer 1 Report

High Thermal Conductive Graphite Films derived from the 2 Graphitization of Chemically Imidized Polyimide Films

The paper can be accepted after the following additions and changes.

1) The accuracy of the measurement technique does not justify two digits after the decimal (e.g. 1764.59). The authors should round the values, and provide discussion of the standard error and accuracy of the measurements. More details of the thermal measurements should be provided in the main text to support the claims of very large thermal conductivity values.  

2) The introduction should have some discussion of the intrinsic thermal conductivity of graphite, graphene (individual planes) and related materials with proper references to review papers. The relevant literature include  Thermal properties of graphene and nanostructured carbon materials,” Nature Mater., 10, 569–581, 2011; “Phonons and thermal transport in graphene and graphene-based materials,” Reports Prog. Phys., 80, 36502, 2017; “Phononics of graphene and related materials,” ACS Nano, 14, pp. 5170-5178, 2020. All these refs contain values for graphite and graphitized materials.

3) How the size and orientation of the continuous layers of graphitized materials affected the thermal conductivity? Some discussion of these issues with references on relevant literature (some in Nanomaterials journal) and results comparison would strengthen the present publication. The relevant papers are: S. Sudhindra, et al., Nanomaterials, 11, 1699, 2021; J. D. Renteria, et al., “Strongly anisotropic thermal conductivity of free-standing reduced graphene oxide films annealed at high temperature,” Adv. Funct. Mater., 25, 4664–4672, 2015.

Author Response

        We sincerely thanks the reviewer for your high evaluation and valuable comments, which is very useful to improve the quality of our papers. We have made the corresponding additions and changes in the manuscript. The following PDF file is a response letter.

Reviewer 2 Report

In the reviewed work  thermal conductive graphite films were obtained by graphitization of polyimide (PI) films with different amounts of chemical catalytic reagents. It has been found that chemically imidized PI (CIPI) films were characterised by higher tensile strength, better thermal stability and higher imidization degree than thermally imidized PI (TIPI) films. The high thermal conductivity of graphite films derived fabricated through graphitization of chemically imidized PI films was attributed to the large in-plane crystallite size and high crystal integrity. The subject of this work is worth to be investigated in detail to speed up  the future development of electronic devices, however, some points need to be addressed in more depth:

-  Line 72: „In this work, several amounts of chemical catalytic reagent are used as catalyst on the imidization reaction of polyamide acid to prepare precursor PI films with different ID and aggregation structures” – please name these catalytic reagents;

- Line 84: „A certain amount of chemical catalytic reagent was added to the polyamide acid (PAA) solution” – what was the amount?

- please explain the role of a catalyst in the imidization reaction of polyamide acid;

- Line 87: „…and heated to 300 ℃ for imidzation reaction” –please provide reaction scheme for this reaction;

- Fig. 4d:  please add egzo/endo direction;

- Table 1 – what is the accuracy of T5% and Tg determination? Have the results with two digits after . (e.g. 369.56) a physical sense?

-  Line 262: „In this work, LED lamps as heat sources are used to evaluate the practical application…” – what is the efficiency and  life time of such a lamp?

- Are the catalyst residues present in the final graphite films?

- Conclusions needs to be re-written; please avoid abstract-like style.

English needs to b checked, e.g. „The graphited films derived from the carbonization and graphitization of CIPI films is conducive”,  „to form orderly structural arrangement, obtain large lattice size”, etc.

Author Response

We sincerely thanks the reviewer for your valuable comments and suggestions. We have made the corresponding revises in the manuscript. Please see the attachment.

Round 2

Reviewer 2 Report

Authors properly explained the issues raised by the reviewer. The revised manuscript can be published as it stands.